# Epsilon Sampling Rocks: Investigating Sampling Strategies for Minimum Bayes Risk Decoding for Machine Translation

**Markus Freitag**
Google Research
freitag@google.com

**Behrooz Ghorbani**[†]
OpenAI
ghorbani@openai.com

**Patrick Fernandes**[‡]
Carnegie Mellon University
Instituto Superior Técnico
pfernand@cs.cmu.edu

## Abstract

Recent advances in machine translation (MT) have shown that Minimum Bayes Risk (MBR) decoding can be a powerful alternative to beam search decoding, especially when combined with neural-based utility functions. However, the performance of MBR decoding depends heavily on how and how many candidates are sampled from the model. In this paper, we explore how different sampling approaches for generating candidate lists for MBR decoding affect performance. We evaluate popular sampling approaches, such as ancestral, nucleus, and top-k sampling. Based on our insights into their limitations, we experiment with the recently proposed **epsilon-sampling** (Hewitt et al., 2022) approach, which prunes away all tokens with a probability smaller than epsilon, ensuring that each token in a sample receives a fair probability mass. Through extensive human evaluations, we demonstrate that MBR decoding based on epsilon-sampling significantly outperforms not only beam search decoding, but also MBR decoding with all other tested sampling methods across four language pairs.

## 1 Introduction

MBR decoding has recently gained attention in Machine Translation (MT) as a decision rule with the potential to overcome some of the biases of beam search decoding in NMT (Eikema and Aziz, 2020; Müller and Sennrich, 2021; Eikema and Aziz, 2021; Freitag et al., 2022a; Fernandes et al., 2022). While most prior work on MBR decoding for MT is based on k-best lists obtained via beam search, Eikema and Aziz (2020) proposed to use an approximation of MBR decoding based on unbiased sampling to overcome the shortcomings of MAP decoding. They demonstrated that samples from the NMT model are faithful to the training data statistics, while beam search is not. Freitag et al. (2022a)

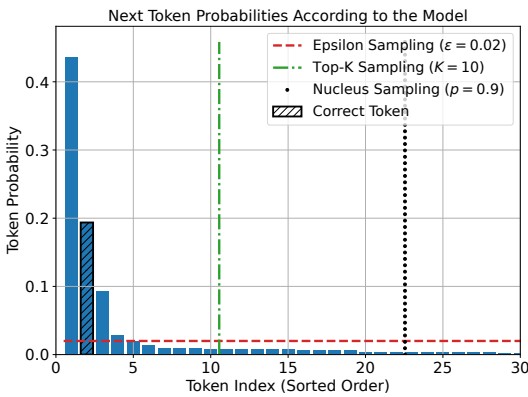

Figure 1: Sorted next token prediction probabilities for an example sentence from newstest2021 English→German. For simplicity, we plot only the top 30 tokens (out of 32k). The correct token is the second most likely token (with $19.3\%$ probability). All but five tokens have a probability of less than $0.02$. However, in aggregate, these low-probability tokens have $21.4\%$ probability.

experimented with different utility functions and showed that the sampling-based MBR decoding approach works well with neural metrics that are fine-tuned on human judgment, such as BLEURT and COMET (Sellam et al., 2020; Rei et al., 2020), significantly outperforming beam search decoding in an expert-based human evaluation.

In this work, we continue this exploration while focusing on the sampling approach used for generating the candidate lists for MBR decoding. We compare MBR decoding using BLEURT on popular sampling approaches such as ancestral sampling, nucleus sampling, or k-best sampling and analyze their advantages and disadvantages. Based on these insights, we explore MBR with the recently-proposed *epsilon-sampling* (Hewitt et al., 2022) approach, which instead of considering only tokens that fall within an aggregated probability mass (nucleus sampling) or a fixed amount of tokens (k-best sampling), prunes away all tokens with a probabil-

---

[†] Work done while working at Google
[‡] Work done while a Student Researcher at Google

ity smaller than epsilon. By doing so, it ensures that every token in each sample gets a decent amount of probability mass from the model. This is not always the case when sampling with nucleus or top-k sampling and the incorrect samples can hurt MBR decoding performance especially for long sequences. In addition, we also explore the relationship between the underlying sampling approach and the sampling's *temperature*.

Our contributions can be summarized as follows:

- We conduct experiments with MBR decoding based on candidate list generated by ancestral, nucleus, top-k and epsilon sampling.

- We run extensive human evaluations directly verifying the effects of different sampling methods on the model generation quality.

- We demonstrate that MBR decoding based on epsilon sampling significantly outperforms not only beam search, but also MBR decoding on all other tested sampling approaches on 4 tested language pairs.

- We conduct expert-based MQM evaluation to verify the quality of our translations.

## 2 Methods

### 2.1 Sampling Approaches

In this section, we provide a brief overview of sampling methods considered in this study. In our analysis, we denote random variables with bold upper-case letters and their realizations with lower-case plain letters. Let $P_{\text{model}}(\mathbf{Y}_t = y_t | \mathbf{X} = x, \mathbf{Y}_{1:t-1} = y_{1:t-1})$ denote the probability assigned by the model to token $y_t$ at time $t$, conditioned on the source ($\mathbf{X} = x$) and the target tokens generated so far ($\mathbf{Y}_{1:t-1} = y_{1:t-1}$). To simplify the notation, when there is no room for misinterpretation, we often omit the random variables and simply write $P_{\text{model}}(y_t | x, y_{1:t-1})$. We consider the following strategies for generating samples from the model:

**Ancestral Sampling:** Ancestral sampling simply draws $y_t$ from $P_{\text{model}}(y_t | x, y_{1:t-1})^{1/\tau}$. Here, $\tau$ is the sampling temperature hyper-parameter which determines the peakedness of the distribution. While this approach is simple and faithful to the model, it is highly sensitive to biases and errors present in the estimated model distribution $P_{\text{model}}$. In particular, the tail of the distribution

(low-probability tokens) is often believed to be highly unreliable. Other sampling strategies attempt to alleviate this issue by systematically trimming the tail.

**Top-k Sampling:** Top-k sampling is a simple modification of ancestral sampling that steers the generation towards high probability tokens. Let $S_{t,k}$ be the set corresponding to $k$ highest probability tokens at time $t$. Then top-k sampling chooses token $y$ at time $t$ with probability proportional to

$$\begin{cases} P_{\text{model}}(y|x, y_{1:t-1})^{1/\tau} & \text{if } y \in S_{t,k}, \\ 0 & \text{otherwise.} \end{cases}$$

**Nucleus Sampling:** Similar to top-k sampling, nucleus sampling (Holtzman et al., 2019) also steers the generation away from the lower trail of the model distribution. Let $Q_{t,p}$ be the smallest possible set of tokens that covers a fraction $p$ of the posterior model probability at time $t$. Then nucleus sampling chooses token $y$ at time $t$ with probability proportional to

$$\begin{cases} P_{\text{model}}(y|x, y_{1:t-1})^{1/\tau} & \text{if } y \in Q_{t,p}, \\ 0 & \text{otherwise.} \end{cases}$$

As $Q_{t,p}$ is the smallest possible set covering the predefined fraction $p$, it includes only the upper tail of the distribution and discards $1 - p$ fraction of the tokens from the lower trail.

**Epsilon Sampling:** Epsilon sampling (Hewitt et al., 2022) employs a simple, yet effective, strategy for pruning unreliable, low-probability tokens. Let $\epsilon \leq 1$ be a non-negative threshold. Epsilon sampling chooses token $y$ at time $t$ with probability proportional to

$$\begin{cases} P_{\text{model}}(y|x, y_{1:t-1})^{1/\tau} & P_{\text{model}}(y|x, y_{1:t-1}) \geq \epsilon, \\ 0 & \text{otherwise.} \end{cases}$$

In their study, Hewitt et al. (2022) argue that epsilon sampling breaks the relative probability principle. E.g. the prompt *The* should allow many valid continuations. This is particularly true for open-ended generation tasks (e.g. story generation), where there are many possible continuations for a given prompt. In machine translation, however, epsilon sampling can be a useful technique. This is because machine translation is a conditional generation task, where the output is conditioned on the input. This means that there are only a limited number of possible continuations for a given input.

## 2.2 Minimum Bayes Risk Decoding

MBR, or Minimum Bayes Risk, is a decoding algorithm that relies on two essential components: a (translation) model and a utility metric. The translation model $P_{\text{model}}(y|x)$ estimates the probability of any target segment $y$ given a source segment $x$. The utility metric $u(h, r)$ estimates quality of a candidate translation $h$ given a reference translation $r$.

Given a set of hypotheses $\mathcal{H}$, we would like to select the best hypothesis according to its expected utility with respect to the distribution over human references in the space of all sequences $\Omega$, i.e.

$$
\begin{aligned}
h^{\text{best}} &= \arg\max_{h \in \mathcal{H}} \mathbb{E}_{r \sim P_{\text{human}}(\cdot|x)}[u(h, r)] \quad (1) \\
&= \arg\max_{h \in \mathcal{H}} \sum_{r \in \Omega} u(h, r) P_{\text{human}}(r|x).
\end{aligned}
$$

Since $P_{\text{human}}(r|x)$ is unknown, we need to rely on the model estimate instead, i.e.

$$
h^{\text{model}} = \arg\max_{h \in \mathcal{H}} \sum_{y \in \Omega} u(h, y) P_{\text{model}}(y|x) \quad (2)
$$

This substitution assumes that the model provides a good approximation for the true underlying (human translation) distribution. As $\Omega$, the space of all sequences, is infinite, it is impossible to integrate over it, and so MBR relies on Monte-Carlo (MC) estimation, using a finite number of pseudo references $\mathcal{H}_{\text{model}}$ sampled from the model $P_{\text{model}}(\cdot|x)$. This yields,

$$
h^{\text{MBR}} = \arg\max_{h \in \mathcal{H}} \frac{1}{|\mathcal{H}_{\text{model}}|} \sum_{y \in \mathcal{H}_{\text{model}}} u(h, y). \quad (3)
$$

Commonly, one relies on the same set of model hypotheses for $\mathcal{H}$ (candidate pool) and $\mathcal{H}_{\text{model}}$ (pseudo-references), i.e. $\mathcal{H} = \mathcal{H}_{\text{model}}$. In that case, growing $\mathcal{H}_{\text{model}}$ has two beneficial effects: a larger set provides a better approximation of the expected utility (reducing finite sample variance) while the maximum over a finite candidate pool obviously increases as the candidate pool grows.

Growing $\mathcal{H}_{\text{model}}$ is however computationally costly, both to obtain hypotheses and to evaluate their cross-utility. In all our experiments, we adopt the sampling-based approximation to MBR decoding (Eikema and Aziz, 2020) to generate a finite set of samples from a neural machine translation model. Eikema and Aziz (2020) showed that unbiased sampling provides a good approximation for the underlying model distribution. The cost of sampling is linear in the size of the set. Cross-utility can involve evaluating a large neural network as well and the cost of utility computation is generally quadratic in the size of the number of samples. It is important to add that we generate independent samples which implies that sentences with higher model probabilities have a higher chance to be drawn several times. By doing so and not deduping the candidate lists, we do not need to incorporate (again) the model probabilities during MBR decoding.

## 3 Experimental Setup

### 3.1 Data and Model

We run experiments on four language pairs: English↔German (En↔De) and English↔Chinese (En↔Zh). We use an internal web-crawled dataset for training our models. We filter out noisy examples with contrastive data selection as proposed by Wang et al. (2018). After filtering, we have 625 million training examples for En↔De language pair and $1.32$ billion training examples for En↔Zh. We use newstest2019 as our dev set to pick checkpoints and newstest2021 (Akhbardeh et al., 2021) as our test set.

### 3.2 Model

Our translation models are 581M parameter transformer models (Vaswani et al., 2017) with 12 encoder and 12 decoder layers, model dimension size of 1,024, hidden dimension size of 4,096, and the number of multi-attention heads is 16. Our models use a vocabulary of 32k subword units (Kudo and Richardson, 2018) trained separately for each language pair on the parallel data. We train the models until convergences for around 500,000 updates with a batch size of 0.5M tokens. We follow the suggestion of Eikema and Aziz (2020) and train our models without label smoothing.

We run beam search with beam size of 4 (larger beam sizes did not improve quality) and length penalty as described in Equation 10 in Wu et al. (2016) using $\alpha$=0.5. We do not use coverage penalty as this does not improve the results.

For MBR decoding, we generate 1,024 samples for each source sentence and use BLEURT (Sellam et al., 2020) as utility function across the paper.

### 3.3 Human Evaluation

We hired professional translators (7 for En→De, 5 for De→En, 4 for Zh→En, and 4 for En→Zh) and measured translation quality with a document context version of MQM (Lommel et al., 2014) which mimics the setup proposed in Freitag et al. (2021). This includes using the same error categories, severity levels and error weighting schema. As suggested in the study, we weight each major error with 5 and each minor error with 1, except for minor punctuation errors which get a score of 0.1. The final segment-level score is an average over scores from all annotators. We refer the reader to Freitag et al. (2021) for the details on error categories and annotator instructions.

## 4 Experimental Results

To understand why epsilon sampling is the most suitable sampling approach for machine translation, we first investigate the limitations of the commonly used sampling approaches in Section 4.1. We then explore different hyperparameter settings for all sampling approaches in Section 4.2. Since it is not feasible to run human assessment on all hyperparameter settings, we select a subset of the highest quality settings based on BLEURT and conduct an expert-based human evaluation to assess the final quality of this subset in Section 4.3.

### 4.1 Distributional Properties of Sampling Approaches

NMT models usually generate a *dense* distributions over the target vocabulary for each time step. Each token is assigned a non-zero probability, even when it is completely unrelated to the source query. When combined with large vocabularies, this means that a considerable (cumulative) probability mass is assigned to undesirable tokens at the *tail* of the distribution. Instead of considering the full vocabulary as done in ancestral sampling, nucleus and top-k sampling are designed to mitigate this problem by *trimming* the tail of the distribution (Figure 1). Even though both nucleus and top-k sampling prune away a significant number of tokens, they potentially still consider tokens that have very low probability according to the model distribution.

During top-k sampling, the number of non-trimmed tokens is independent of the model *confidence* (entropy): this means that for cases where the model is too confident, it might fail to prune a

large number of undesirable tokens, while for cases where the model is uncertain, it might prune a large set of valid tokens (Figure 2).

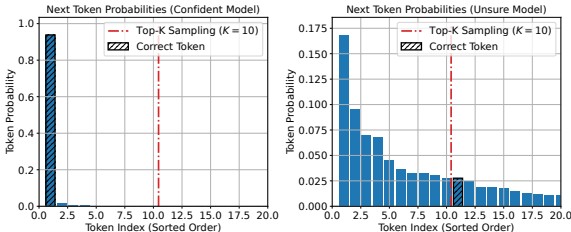

Figure 2: Top-K sampling is insensitive to model confidence.

Nucleus sampling considers the most probable tokens within an accumulated probability mass of $p$. Even with smaller $p$ values, we sometimes need to consider several hundreds of tokens. Figure 3 shows an example where we need almost 400 tokens to get an accumulated probability mass of 0.9. Many of the 400 tokens have very low probability on their own, but as their accumulated mass is not small, there is a large chance that one of the low probability tokens will be considered by nucleus sampling.

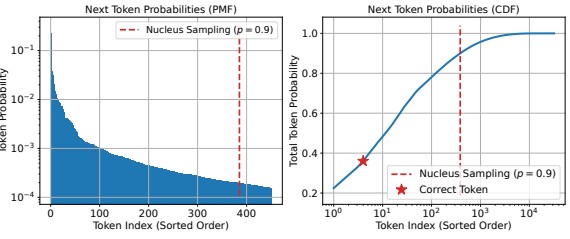

Figure 3: Nucleus sampling can have unpredictable behavior and lead to many low-probability tokens being considered.

We believe that if tokens have a very low probability according to the model, we shouldn't consider them at all. This is precisely the motivation behind using epsilon sampling. By setting an adequate threshold $\epsilon$, we can exclude the undesirable low-probability tokens. This leads to a pruned set of tokens that has both variable number of tokens and variable accumulated probability mass.

We want to highlight another interesting comparison of the different sampling approaches in Figure 4. The cumulative probability mass is larger for epsilon sampling when sampling 1024 times from the model while the other sampling approaches converge to the same area. This might indicate that we can generate more diverse translations when us-

ing epsilon sampling and, importantly, this means we can get better estimations of the true utility in Equation 3.

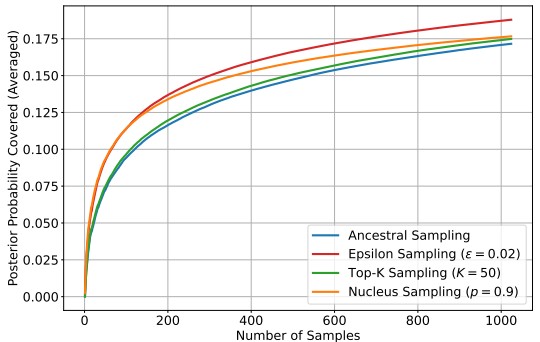

Figure 4: (Cumulative) probability mass *over sentences*, covered by each sampling method as we increase the number of samples.

## 4.2 Hyperparameter Search

All of the sampling approaches introduced in Section 2.1 have one or two hyperparameters. We ran a large number of experiments on English→German and used this language pair as our development language pair for hyperparameter selection. We then used the exact same hyperparameters for other language pairs to investigate how well they generalize. All MBR decoding results used BLEURT as the utility function. For each sampling approach, we picked 1-2 hyperparameter settings purely based on BLEURT. In the final evaluation, we will investigate how well the BLEURT scores correlate with human assessment when used both as a utility function and an evaluation metric.

### 4.2.1 Ancestral Sampling

Ancestral sampling has one hyperparameter, the temperature $\tau$. We ran MBR decoding on candidate lists generated via ancestral sampling with $\tau = 0.65, 0.75, 0.85, 1.0, 1.1$ and compared its performance to the translations generated with beam search decoding. In addition to just looking at the performance of MBR decoding using all $1024$ samples, we also looked at how well MBR decoding performs when reducing the candidate list size.

In Figure 5, we observe that using a lower temperature ($\tau$) is favorable when the candidate size is small. This is because a lower temperature encourages the model to generate more likely tokens, which is important when the candidate list is small and there are only a few possible translations to

choose from. For larger candidate sizes, higher temperatures perform better as we can be more risky and thus also consider interesting translations with lower model probability, but potentially even higher quality. In this paper, we will focus on large candidate sizes and thus choose $\tau$=1.0 for our experiments going forward.

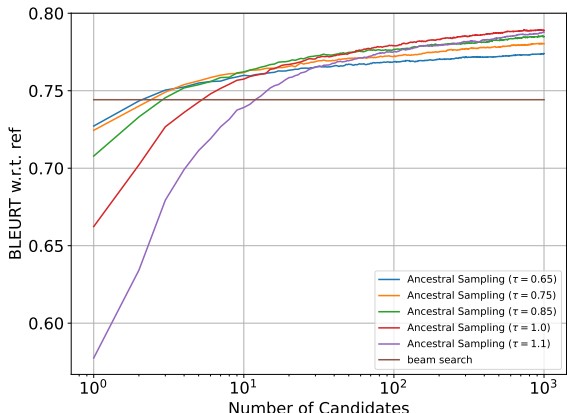

Figure 5: Ancestral Sampling for newstest2021 English→German for various temperatures.

### 4.2.2 Top-K Sampling

Top-k sampling has two parameters: $\tau$ and $k$. Figure 6 shows different hyperparameter settings. Similar to ancestral sampling, using a lower temperature helps when the candidate size is small. Once the candidate size is large, the quality of the different hyperparameter settings are almost identical. We decided to use the more traditional settings ($k$=10, $\tau$=1.0), and ($k$=50, $\tau$=1.0) for our experiments going forward.

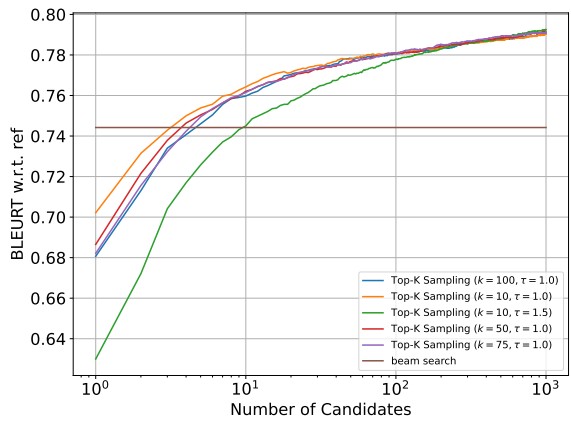

Figure 6: Top-K Sampling for newstest2021 English→German for different settings.

### 4.2.3 Nucleus Sampling

Nucleus sampling has two hyperparameters: $\tau$ and $p$. Figure 7 illustrates the performance of different hyperparameter settings. First, we observe the same behavior that we saw with ancestral and top-k sampling: using a lower temperature is favorable when the candidate size is small. This is because a lower temperature encourages the model to generate more likely tokens, which is important when the candidate list is small and there are only a few possible translations to choose from. Second, we find that using a higher temperature ($\tau$=1.5) combined with $p$=0.9 outperforms all other tested settings when using a large (=1024) candidate list. This is because a higher temperature allows the model to generate more creative and interesting translations, while p=0.8 ensures that the model does not generate too many low-probability tokens. Therefore, we will use this setting ($p$=0.9, $\tau$=1.5) in our final evaluation.

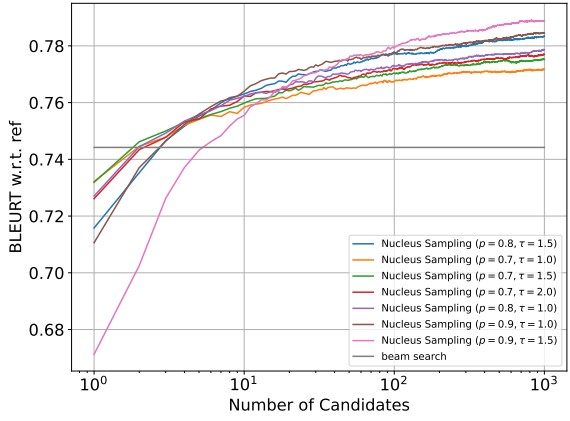

Figure 7: Nucleus Sampling for newstest2021 English→German for different settings.

### 4.2.4 Epsilon Sampling

Epsilon sampling has two parameters: $\tau$ and $\epsilon$. Figure 8 shows the performance of different parameter settings with MBR decoding. Similar to the other sampling strategies, a low temperature ($\tau$) is needed to yield good translation quality for small candidate sizes. Increasing the temperature results in higher BLEURT scores when using a large candidate list.

We have to highlight that very high temperatures ($\tau \geq 2$) yield a sharp drop in translation quality when used for the traditional sampling approaches[1],

---

[1]Ancestral, top-k and nucleus sampling with $\tau$=2.0 yield 0.45, 0.65 and 0.7 BLEURT for candidate list size of 1024.

but show promising results for epsilon sampling. The limitations of traditional sampling approaches, as discussed in Section 4.1, are that they can consider very low-probability tokens especially when combined with a large temperature. This can lead to a sharp drop in translation quality.

We chose two settings for our final evaluation: The highest scoring setting ($\tau$=2.0, $\epsilon$=0.02) and ($\tau$=1.0, $\epsilon$=0.02) with the same epsilon threshold, but lower temperature to measure the direct impact of a large temperature during epsilon sampling.

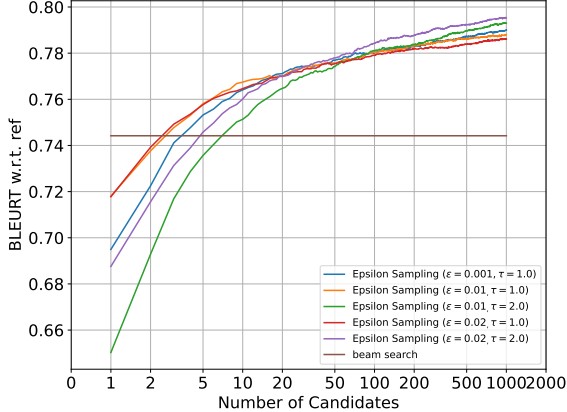

Figure 8: Epsilon Sampling for newstest2021 English→German for different settings.

### 4.3 Final Evaluation

The experimental results of MBR decoded translations with 1024 samples comparing the six sampling settings chosen in Section 4.2 including an expert-based MQM human evaluation are summarized in Table 1. Our findings are consistent across all four tested language pairs. Based on the expert-based MQM human evaluation, we come to the following conclusions:

- MBR decoding with BLEURT outperforms beam search decoding across all four language pairs independent of the underlying sampling approach.

- For all language pairs, MBR Epsilon ($\epsilon$=0.02, $\tau$=1.0) sampling outperforms all other MBR runs.

- Higher temperature in epsilon sampling can lead to "reward over-fitting": even though $\tau$=2 generally has higher BLEURT scores (the utility function) for most language-pairs, humans tend to prefer translation obtained from $\tau$=1. This indicates that that high temperature might

|  | | Automatic Evaluation | | | | Model | Human Eval |
|---|---|---|---|---|---|---|---|
|  | | BLEU | CHRF | BLEURT | COMET20 | logP | MQM ↓ |
| Human Transl. (Ref-D) | | 31.5 | 60.9 | 76.1 | 59.0 | -30.6 | 0.55 |
| MBR | Epsilon ($\epsilon$=0.02, $\tau$=1.0) | 34.7 | 63.6 | 78.6 | **61.9** | -10.8 | **1.04** |
|  | Top-k ($k$=10, $\tau$=1.0)† | 33.5 | 63.0 | 79.0 | **61.9** | -13.6 | 1.13 |
|  | Top-k ($k$=50, $\tau$=1.0)† | 32.6 | 62.3 | 79.1 | 61.3 | -15.6 | 1.15 |
|  | Ancestral ($\tau$=1.0)† | 33.3 | 62.7 | 78.9 | 61.1 | -15.0 | 1.16 |
|  | Nucleus ($p$=0.9, $\tau$=1.5)† | 31.9 | 61.9 | 78.9 | 60.8 | -15.8 | 1.25 |
|  | Epsilon ($\epsilon$=0.02, $\tau$=2.0)† | 27.5 | 59.0 | **79.6** | 60.0 | -20.5 | 1.26 |
| Beam 4† | | 37.1 | 65.1 | 74.4 | 58.0 | -5.4 | 1.36 |

(a) English→German (*Ref-C*)

|  | | Automatic Evaluation | | | | Model | Human Eval |
|---|---|---|---|---|---|---|---|
|  | | BLEU | CHRF | BLEURT | COMET20 | logP | MQM ↓ |
| Human Transl. (Ref-A) | | 28.0 | 28.2 | 63.5 | 42.7 | -54.2 | 1.86 |
| MBR | Epsilon ($\epsilon$=0.02, $\tau$=1.0) | 29.6 | 26.9 | 67.2 | 44.1 | -13.4 | **3.61** |
|  | Top-k ($k$=10, $\tau$=1.0)† | 29.3 | 26.8 | 67.8 | 44.0 | -16.2 | 3.97 |
|  | Nucleus ($p$=0.9, $\tau$=1.5)† | 28.2 | 26.1 | 68.6 | **45.6** | -20.6 | 4.09 |
|  | Top-k ($k$=50, $\tau$=1.0)† | 28.7 | 26.3 | 68.5 | 45.1 | -18.9 | 4.10 |
|  | Ancestral ($\tau$=1.0)† | 28.4 | 26.1 | 68.5 | 44.9 | -20.0 | 4.12 |
|  | Epsilon ($\epsilon$=0.02, $\tau$=2.0)† | 26.6 | 25.0 | **68.9** | 45.1 | -23.4 | 4.40 |
| Beam 4† | | 29.6 | 28.5 | 61.9 | 36.0 | -7.6 | 4.77 |

(b) English→Chinese (*Ref-B*)

|  | | Automatic Evaluation | | | | Model | Human Eval |
|---|---|---|---|---|---|---|---|
|  | | BLEU | CHRF | BLEURT | COMET20 | logP | MQM ↓ |
| MBR | Epsilon ($\epsilon$=0.02, $\tau$=1.0) | 33.1 | 60.9 | 76.1 | 62.8 | -6.9 | **1.07** |
|  | Nucleus ($p$=0.9, $\tau$=1.5)† | 31.0 | 59.4 | 76.3 | 61.3 | -9.4 | 1.22 |
|  | Epsilon ($\epsilon$=0.02, $\tau$=2.0)† | 28.2 | 57.8 | **76.4** | 60.5 | -11.4 | 1.25 |
| Human Transl. (Ref-B)† | | 29.5 | 57.7 | 73.5 | 56.4 | -26.7 | 1.26 |
| MBR | Top-k ($k$=10, $\tau$=1.0)† | 32.1 | 60.4 | 76.3 | 62.4 | -7.8 | 1.31 |
|  | Top-k ($k$=50, $\tau$=1.0)† | 31.9 | 60.2 | 76.3 | 62.2 | -8.5 | 1.38 |
|  | Ancestral ($\tau$=1.0)† | 32.0 | 60.4 | 76.2 | 61.8 | -8.7 | 1.40 |
| Beam 4† | | **34.7** | 62.4 | 74.5 | 61.1 | -4.7 | 1.43 |

(c) German→English (*Ref-A*)

|  | | Automatic Evaluation | | | | Model | Human Eval |
|---|---|---|---|---|---|---|---|
|  | | BLEU | CHRF | BLEURT | COMET20 | logP | MQM ↓ |
| Human Transl. (Ref-B) | | 28.2 | 59.6 | 69.6 | 47.3 | -53.1 | 1.53 |
| MBR | Epsilon ($\epsilon$=0.02, $\tau$=1.0) | 26.4 | 56.4 | 70.2 | 43.4 | -18.3 | **3.02** |
|  | Top-k ($k$=10, $\tau$=1.0) | 25.7 | 55.9 | **70.4** | 43.2 | -21.1 | 3.12 |
|  | Top-k ($k$=50, $\tau$=1.0)† | 25.3 | 55.5 | 70.2 | 42.1 | -23.1 | 3.29 |
|  | Ancestral ($t$=1.0)† | 25.2 | 55.5 | 70.1 | 41.8 | -24.2 | 3.49 |
|  | Epsilon ($\epsilon$=0.02, $\tau$=2.0)† | 22.3 | 53.8 | 70.1 | 41.8 | -28.4 | 3.38 |
|  | Nucleus ($p$=0.9, $\tau$=1.5)† | 24.1 | 54.5 | 69.8 | 39.9 | -27.4 | 3.57 |
| Beam 4† | | 27.2 | 54.5 | 65.8 | 30.2 | -10.5 | 3.61 |

(d) Chinese→English (*Ref-A*)

Table 1: Actual utility, log-likelihood (logP) and MQM score for different MBR methods and beam search on newstest2021. All MQM results labelled with † are significantly worse than MBR Epsilon ($\epsilon$=0.02, $\tau$=1.0) on PERM-BOTH significance testing (Deutsch et al., 2021) with p=0.001.

be exploiting weaknesses in the utility (Amrhein and Sennrich, 2022).

- Beam search consistently outperforms MBR decoding when combined with any sampling approach, when measuring by BLEU. This highlights the low correlation of BLEU with human judgement as discussed in several publications before (Freitag et al., 2022b).

- As observed before, the human translation for newstest2021 German→English contains many errors and is already outperformed by MBR decoding. This highlights again the importance of acquiring high level human translations when comparing humans with machines.

## 5 Related Work

Minimum Bayes Risk (MBR) decoding stems from statistical decision theory from the principal of maximisation of expected utility (Bickel and Doksum, 1977; Berger, 1985). MBR has been applied to parsing (Goodman, 1996; Sima'an, 2003) and speech recognition (Stolcke et al., 1997; Goel and Byrne, 2000). The same idea was later applied to bilingual word alignment (Kumar and Byrne, 2002) and machine translation (Kumar and Byrne, 2004). MBR was used to maximize overlap metrics such as BLEU (Papineni et al., 2002) with statistical MT systems (Kumar and Byrne, 2004; Smith and Eisner, 2006; Tromble et al., 2008).

After the advent of neural machine translation, most methods relied on beam search to approximate MAP decoding (Bahdanau et al., 2015; Gehring et al., 2017; Vaswani et al., 2017). MBR decoding has recently gained attention in MT as a decision rule with the potential to overcome some of the biases of MAP decoding in NMT (Eikema and Aziz, 2020; Müller and Sennrich, 2021; Eikema and Aziz, 2021). While most prior work on MBR decoding for MT is based on k-best lists obtained via beam search, Eikema and Aziz (2020) proposed to use an approximation of MBR decoding based on unbiased sampling to overcome the shortcomings of MAP decoding. They demonstrated that samples from the NMT model are faithful to the training data statistics, while beam search is not. We adopt their sampling-based MBR decoding approximation in all our experiments. Freitag et al. (2022a); Fernandes et al. (2022) further explored MBR using neural-based utility functions.

They demonstrated that neural-based utility function like BLEURT and COMET outperform lexical overlap metrics. Further, Fernandes et al. (2022) found that alternatives to *ancestral* sampling could lead to improved performance of MBR-based decoding. (Amrhein and Sennrich, 2022) found that, despite promising results with neural metrics in machine translation evaluation, MBR might exploit biases towards faulty translations with high scores that exist in these metrics (reward over-fitting).

MBR has since been extensively in used in submissions to various machine and speech translation *shared tasks* with good results (Nowakowski et al., 2022; Jon et al., 2022; Yan et al., 2022), showcasing its potential to improve translation quality.

Besides the approaches discussed in Section 2.1, other sampling approaches that attempt to fix some of the issues with vanilla sampling have been proposed. For example, (Meister et al., 2022) introduced the concept of *typical sampling*, which proposed pruning tokens whose probability deviates alot from model's (conditional) *entropy* (so potentially both high and low probability tokens), and show this reduces degenerate repetitions and improves generation quality.

## 6 Conclusion

In this paper, we investigated the impact of different sampling approaches (ancestral, nucleus, top-k, and epsilon sampling) during MBR decoding. We analysed the limitations of the traditional sampling approaches and proposed combining the recently proposed *epsilon-sampling* with MBR. Finally, we conducted human evaluations on four language pairs, and showed that MBR decoding based on epsilon-sampling significantly outperforms not only beam search but also MBR decoding with all other tested sampling approaches. We believe that the results of this study are significant for several reasons. First, they demonstrate that epsilon-sampling is a promising approach for improving the quality of MBR decoding. Second, they provide insights into the relative performance of different sampling approaches, suggesting that epsilon sampling is a promising direction even outside of MBR decoding and for other non open-ended generation tasks.

In future work, we plan to conduct further human assessment to investigate the impact of using a smaller candidate size on the performance of MBR decoding.

## Limitations

Automatic evaluation of translations generated with MBR decoding can be challenging when the quality metric (BLEURT, COMET) and the utility function (in our case BLEURT) are the same or at least related. This is because there may be *metric honey pots* (Freitag et al., 2020) that yield high metric scores, but are not necessarily of high quality. MBR decoding is particularly susceptible to this issue, as it is designed to find the translation with the highest possible metric score.

To address the problem of potential overfitting of the metric, it is necessary to run a final human assessment to verify the quality of the translations. However, human assessment is expensive and time-consuming, so it is not possible to assess all translations generated as part of this study. We had to limit ourselves to a subset of translations that we could send out for human assessment. It is possible that some of the translations that we did not assess are of higher quality than the translations that we did assess.

Another limitation of MBR decoding with neural metrics is its decoding speed. In our experiments, we had to generate 1024 samples for each source sentence and each hyperparameter setting, and then score each pair of samples with BLEURT. The cost of sampling is linear in the size of the set, and the cost of utility computation is generally quadratic in the size of the number of samples. This makes MBR decoding in its current form very expensive and impractical for most use cases.

## Ethics Statement

The annotators were compensated fairly and did not have to disclose any personal information during the annotation process. All of the test sets used in this study are publicly available, and annotators were allowed to label sensitive information if necessary.

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

## A  Token-Level Probabilities

We believe that the strong results of MBR decoding based on epsilon sampling are mainly due to the epsilon threshold, which avoids translations that contain at least one token with a low (smaller than epsilon) token-level probability. By adding this epsilon guard, we avoid translations where not every token is backed up by at least some decent probability mass from the model. To investigate this hypothesis, we looked at the MQM human annotations, in particular at the major error spans annotated for the other sampling strategies, to connect error spans with token-level probabilities. Some example translations for German→English can be seen in Table 2. Error spans as labeled by professional translators are highlighted in red. Interestingly, we can see that all major error spans contain at least one token that has a token-level probability smaller than 0.02. Consequently, none of the translations generated by MBR decoding based on either top-k or nucleus sampling are in the candidate pool when doing MBR decoding with epsilon sampling.

| | |
|---|---|
| Source | Ein Sicherheitsdienst überwacht das Ausgehverbot. |
| Reference | A security service monitors the curfew. |
| Epsilon ($\epsilon$=0.02, $\tau$=1.0) | A security guard supervises the curfew. |
| Top-k (k=10, t=1.0) | A security guard keeps watch over the area. |
| Nucleus (p=0.9, t=1.5) | A security guard keeps watch over the premises. |

(a) Example 1: The major error spans *area* and *premises* have token-level probabilities smaller than 0.02.

| | |
|---|---|
| Source | Bundesarbeitsminister Hubertus Heil (SPD) will nach den nun geplanten strengen Vorschriften gegen Missstände in der Fleischindustrie auch andere Branchen überprüfen. |
| Reference | The Federal Minister of Labor Hubertus Heil (SPD) wants to investigate other sectors after the currently planned, strict measures against abuses in the meat industry. |
| Epsilon ($\epsilon$=0.02, $\tau$=1.0) | Federal Labor Minister Hubertus Heil (SPD) also wants to check other industries according to the now planned strict regulations against abuses in the meat industry. |
| Top-k (k=10, t=1.0) | Federal Labor Minister Hubertus Heil (SPD) wants to examine other industries following the new, strict regulations against abuses in the meat industry. |
| Nucleus (p=0.9, t=1.5) | German Labor Minister Hubertus Heil (SPD) plans to audit other sectors as well, based on the new strict rules that would combat abuses in the meat industry. |

(b) Example 2: The token *new* has a token-level probability smaller than 0.02 in both error spans.

| | |
|---|---|
| Source | Beruflich angekommen - privat noch nicht |
| Reference | Professionally achieved - privately not there yet. |
| Epsilon ($\epsilon$=0.02, $\tau$=1.0) | Arrived professionally - not privately yet |
| Top-k (k=10, t=1.0) | Arrived at professional level - privately not yet . |
| Nucleus (p=0.9, t=1.5) | Newly arrived in professional life - but not in private life yet |

(c) Example 3: The first token in all errors spans (*Newly*, *professional*, and .) have token-level probabilities smaller than 0.02.

Table 2: Example translations where we avoid major translation errors by using epsilon sampling instead of nucleus or top-k sampling during MBR decoding. Major translation errors are highlighted in red and were labeled by human annotators as part of our MQM evaluation. Interestingly, all of these major errors were avoided when switching to epsilon sampling, as the error spans contain at least one token with a token-level probability smaller than 0.02 (the epsilon threshold used in our experiments). In all of the examples, the translations generated with MBR decoding based on epsilon sampling are error-free.