# OpenReview forum: "Epsilon Sampling Rocks: Investigating Sampling Strategies for Minimum Bayes Risk Decoding for Machine Translation"
_EMNLP/2023/Conference — EMNLP 2023 Findings_

### Official Review · Reviewer_FrLm · 2023-08-02

**Soundness:** 4

**Excitement:**

4: Strong: This paper deepens the understanding of some phenomenon or lowers the barriers to an existing research direction.

**Paper Topic And Main Contributions:**

This paper investigates different sampling strategies for generating candidate lists for Minimum Bayes Risk (MBR) decoding for neural machine translation. The performance of MBR decoding depends on how candidates are sampled from the model. The authors evaluate popular sampling strategies, such as ancestral, nucleus, and top-k. To address the limitations, they have proposed the epsilon-sampling approach. This approach prunes all tokens with a probability smaller than a given threshold (epsilon), ensuring that each token in the sample gets a fair share of the probability mass. They argue that this approach benefits long sequences where incorrect samples can negatively affect MBR decoding performance. The paper also discusses the effect of the underlying sampling approach and the sampling's temperature.

To validate their proposal, the authors conduct experiments with MBR decoding based on candidate lists generated by different sampling methods, including ancestral, nucleus, top-k, and epsilon sampling. They run human evaluations to verify the effects of other sampling methods on model generation quality. The authors demonstrate that MBR decoding based on epsilon sampling significantly outperforms not only beam search but also MBR decoding based on all other tested sampling approaches across four tested language pairs. They also conduct the Multidimensional Quality Metrics (MQM) evaluation to verify the quality of their translations.

**Questions For The Authors:**

The authors should address the weaknesses mentioned in the above section.

**Reasons To Accept:**

1. The paper compares popular sampling approaches for generating candidate lists for MBR decoding, including ancestral, nucleus, top-k, and epsilon sampling. It provides insights into the relative strengths and weaknesses of different sampling approaches for MBR decoding.

2. The authors have proposed the epsilon-sampling approach. This approach prunes all tokens with a probability smaller than a given threshold (epsilon), ensuring that each token in the sample gets a fair share of the probability mass. They argue that this approach benefits long sequences where incorrect samples can negatively affect MBR decoding performance. The paper also discusses the effect of the underlying sampling approach and the sampling's temperature.

3. The paper demonstrates through experiments and human evaluations that MBR decoding based on epsilon sampling outperforms not only beam search, but also MBR decoding on all other tested sampling methods. The authors evaluate across 4 language pairs - English <-> German and English <-> Chinese. The paper verifies the quality of the translations through expert-based MQM evaluations.

4. The paper investigates the hyperparameter search for each sampling method, running experiments to determine optimal settings.

5. The paper is well-structured and well-written.

**Reasons To Reject:**

1. The authors highlight that MBR decoding, in its current form, is computationally expensive and time-consuming, making it impractical for most use cases.

2. The experiments were conducted on four language pairs. Including more language pairs could have provided a more comprehensive evaluation of the performance of the different sampling strategies.

3. Although the authors have mentioned the limitations in the paper, they should provide a more detailed plan on how they plan to address these drawbacks in their future work.

4. The authors should provide ablation studies that could isolate the impact of specific factors underlying epsilon sampling's strong performance.

**Reproducibility:**

4: Could mostly reproduce the results, but there may be some variation because of sample variance or minor variations in their interpretation of the protocol or method.

**Reviewer Confidence:**

3: Pretty sure, but there's a chance I missed something. Although I have a good feel for this area in general, I did not carefully check the paper's details, e.g., the math, experimental design, or novelty.

---

> ### Author Rebuttal · Authors · 2023-08-25
>
> Thanks a lot for the review. Here are the answers for your questions:
> Q1: There are many use cases where clients actually prefer to get high quality output and can wait for the translations. That said, epsilon sampling is the first step in the right direction, but we need to do further research to come up with a faster, similar reliable approach.
> Q2: We agree that more language pairs would be nice. However, as we use BLEURT as a utility function in MBR decoding, we cannot rely on BLEURT as the evaluation metric any longer (we basically cherry-pick the translation with the highest expected BLEURT from the candidate list during MBR) and need to conduct a human eval. Human evals are expensive and time consuming and conduction human evals for 4 LPs is way more than what is typically conåducted in an EMNLP paper.
> Q3: We feel that it is not necessary to lay out our next follow up research projects in the paper. We will add some suggestions at the end of the paper that include distilling the utility function and further investigate how we can prune the candidate lists.
> Q4: Can you elaborate a little bit more on this. What do you want to investigate in addition to what we already have? We have more experiments that we would have loved to add to the paper, but are limited by the page limit.

---

### Official Review · Reviewer_b3eK · 2023-08-04

**Typos Grammar Style And Presentation Improvements:** n/a
**Soundness:** 5

**Excitement:**

5: Transformative: This paper is likely to change its subfield or computational linguistics broadly. It should be considered for a best paper award. This paper changes the current understanding of some phenomenon, shows a widely held practice to be erroneous in someway, enables a promising direction of research for a (broad or narrow) topic, or creates an exciting new technique.

**Missing References:**

n/a

**Paper Topic And Main Contributions:**

This paper systematically investigates various popular sampling strategies for MBR decoding. The conclusion is that the epsilon sampling is the most promising one. The background part provides a clear overview of all these sampling methods and the MBR decoding itself. The experiments part is comprehensive and solid, which well supports the main conclusion.

**Questions For The Authors:**

Do you have a suggestion on applying this method in a latency-sensitive situation?

How would you compare this MBR decoding with MAP decoding and reranking?

**Reasons To Accept:**

This paper brings up the MBR decoding to the community which is a nice alternative to the MAP decoding in some cases. Most importantly, it draws a practical conclusion that the epsilon sampling is the best with extensive experiments, which is very informative and helpful for related researchers and practitioners.

**Reasons To Reject:**

Although MBR decoding is costly and won't be ideal to use in many cases, it's not the main concern of the topic of this paper. I don't see a clear reason to reject.

**Reproducibility:**

5: Could easily reproduce the results.

**Reviewer Confidence:**

4: Quite sure. I tried to check the important points carefully. It's unlikely, though conceivable, that I missed something that should affect my ratings.

---

> ### Author Rebuttal · Authors · 2023-08-25
>
> Thanks a lot for your review!
>
> To answer your questions: Q1: This is still an active research area. One big advantage of epsilon sampling is that we can use a smaller candidate list than what we used to use for ancestral and nucleus sampling generated candidate lists. Currently the bottleneck is mainly the heavy computational cost of the utility function. Distilling the utility function into a smaller model seems like a promising direction. Q2: For reranking, we need a QE metric as we do not have access to a reference translation. Based on (https://aclanthology.org/2022.tacl-1.47/), reranking underperforms MBR decoding as it sometimes prefers translations that are unrelated to the source. MBR decoding gives us the security net when compared to all the pseudo references to ensure that the translation is also accurate. MAP and MBR decoding are quite different and their output quality can not only differ, but also the translation style. MAP decoding generates very literal, boring translations (also leading to very high BLEU scores), while MBR decoding generates very natural translations.

---

### Official Review · Reviewer_YyrA · 2023-08-05

**Typos Grammar Style And Presentation Improvements:** N/A
**Soundness:** 4

**Excitement:**

2: Mediocre: This paper makes marginal contributions (vs non-contemporaneous work), so I would rather not see it in the conference.

**Missing References:**

N/A

**Paper Topic And Main Contributions:**

This paper investigates the feasibility of combining the machine translation model decoding with epsilon-sampling method, which help filter out the decoding candidates with quite low probabilities. Based on the experimental results, the proposed method can give better quality of translation outputs than other baselines, like top-k and nucleus sampling. Quantitative analyses also demonstrate the effectiveness of this method. Overall, this paper is well-written and easy-to-follow, and it is well-structured. However, the merits of this paper might be not very sufficient for top-tier conference paper.

**Questions For The Authors:**

1.  I'm afraid the content of this paper might be insufficient for a conference paper. The instantiation of epsilon sampling for machine translation might be too incremental. Could you help show some details of your method, especially the differences between your method and the conventional epsilon sampling (especially for MT)? This can help others quickly understand your thoughts and merits in this research.
2. As the decoding strategy lies in the filtering on probabilites of candidates when decoding, I'm concerned that some other MT approaches which changes the learning objectives with respect to the ground truth probabilities might have different conclusions. For example, for the models whose learning objective excludes label smoothing (which is a common practice when training MT models nowadays), can your method still performs better than other baselines? If so, how would you change the hyper-parameters of your method?

**Reasons To Accept:**

1. An efficient method for decoding translation output using epsilon decoding.
2. Quantitative analyses on the performance of epsilon decoding.
3. Well-written and easy-to-follow.

**Reasons To Reject:**

1. It's concerned that this work might be a simple incremental work (See Q1)
2. A potential risk for this method lies in the generality of MT models for decoding (See Q2)

**Reproducibility:**

3: Could reproduce the results with some difficulty. The settings of parameters are underspecified or subjectively determined; the training/evaluation data are not widely available.

**Reviewer Confidence:**

3: Pretty sure, but there's a chance I missed something. Although I have a good feel for this area in general, I did not carefully check the paper's details, e.g., the math, experimental design, or novelty.

---

> ### Author Rebuttal · Authors · 2023-08-25
>
> Thanks a lot you for your review.
>
> Regarding your question 1: We use epsilon sampling to generate the candidate list for MBR decoding. We do not simply test epsilon sampling for MT. There are a couple of novel contributions which as far as we know have not been investigated:
>
> We compare different sampling approaches for MBR Decoding. There are 2 existing papers about MBR decoding for NMT and one is using ancestral sampling, the other is using nucleus sampling. There was never a study which sampling approach performs better. Also no one ever tried MBR decoding with top-k sampling or epsilon sampling.
> Based on the limitation and failure modes of MBR decoding with the popular sampling approaches, we apply epsilon sampling to generate the candidate lists. The usage of epsilon sampling in MT is not only new, but using it for MBR decoding is novel as well. We also investigate the impact of various temperatures on the candidate list and its impact on MBR decogin.
> The experimental results are pretty good and set a new standard in NMT. We are convinced that this is interesting for the EMNLP community and will have impact beyond MT.
> All experiments and the choice for epsilon sampling are well motivated. In fact, we implemented epsilon sampling before it was published by others as we designed a sampling approach that should fit MBR decoding as much as possible.
> We believe that all the points above are more than sufficient enough to justify an EMNLP paper.
>
> Regarding question 2: This is a very good question and we will add a discussion about this in the paper. Our models are actually trained without label smoothing (l 231) as this would corrupt the probability distribution for MBR decoding. This is actually a larger issue for nucleus and ancestral sampling as many incorrect tokens get some decent probability mass assigned (see Section 2 in our paper). For epsilon sampling, this is not an issue as the epsilon threshold prunes away the tokens that some decent probability mass due to label smoothing. We ran a couple of experiments with and without label smoothing and epsilon sampling is quite robust and you do not have to change the epsilon threshold. This is not the case for nucleus or ancestral sampling where we had to change the temperature to get similar results. Thanks again for pointing out this detail. We will add the discussion and the advantage of epsilon sampling to the paper.

---

### Meta-Review · Area_Chair_ehyE · 2023-09-16

**Recommendation:** 4

**Metareview:**

This paper compares different sampling strategies for MBR decoding, including a number of well-known approaches such as top-k and nucleus sampling, and the more obscure epsilon sampling, originally introduced by Hewitt et al., but dismissed as having a "key failure" in high entropy distributions. The main contribution of the paper is showing that in the constrained setting of MT, epsilon sampling is an effective sampling strategy, and in combination with MBR decoding, performs better than other strategies tested in human evaluation.

Reviewers agree that empirical insight by the paper are valuable, and potentially have practical impacts (currently limited by the ineffiency of MBR). There is more disagreement on the more subjective excitement score. On the one extreme, reviewer YyrA finds that repurposing epsilon sampling for MT is an incremental contribution. On the other extreme, reviewer FrLm gives the paper credit for "proposing the epsilon sampling approach", and b3eK for "bring[ing] up the MBR decoding to the community".

Factually, the view of YyrA is more precise, since epsilon sampling is not introduced in this paper, nor can the paper be accepted on the basis of bringing MBR to the community, since it follows a number of other papers on the issue. Still, given the previous obscurity of epsilon sampling, which was even dismissed in the paper introducing it, the empirical findings in this paper are significant and I expect it to have some impact on the field.

---

### Decision · Program_Chairs · 2023-10-07

**Decision:**

Accept-Findings

**Comment:**

This paper compares different sampling strategies for MBR decoding, including a number of well-known approaches such as top-k and nucleus sampling, and the more obscure epsilon sampling, originally introduced by Hewitt et al., but dismissed as having a "key failure" in high entropy distributions. The main contribution of the paper is showing that in the constrained setting of MT, epsilon sampling is an effective sampling strategy, and in combination with MBR decoding, performs better than other strategies tested in human evaluation.

Reviewers agree that empirical insight by the paper are valuable, and potentially have practical impacts (currently limited by the ineffiency of MBR). There is more disagreement on the more subjective excitement score. On the one extreme, reviewer YyrA finds that repurposing epsilon sampling for MT is an incremental contribution. On the other extreme, reviewer FrLm gives the paper credit for "proposing the epsilon sampling approach", and b3eK for "bring[ing] up the MBR decoding to the community".

Factually, the view of YyrA is more precise, since epsilon sampling is not introduced in this paper, nor can the paper be accepted on the basis of bringing MBR to the community, since it follows a number of other papers on the issue. Still, given the previous obscurity of epsilon sampling, which was even dismissed in the paper introducing it, the empirical findings in this paper are significant and I expect it to have some impact on the field.